# High-Power Coupled Wideband Low-Frequency Antenna Design for Enhanced Long-Range Loran-C Timing Synchronization

**DOI:** 10.3390/s25144352

**Published:** 2025-07-11

**Authors:** Jingqi Wu, Xueyun Wang, Juncheng Liu, Chenyang Fan, Chenxi Zhang, Zilun Zeng, Liwei Wang, Jianchun Xu

**Affiliations:** 1State Key Laboratory of Information Photonics and Optical Communications, School of Physical Science and Technology, Beijing University of Posts and Telecommunications, Beijing 100876, China; 2024010635@bupt.cn (J.W.); fanchenyang@bupt.edu.cn (C.F.); zhangchenxi@bupt.edu.cn (C.Z.); zzlzz@bupt.edu.cn (Z.Z.); wangliwei@bupt.edu.cn (L.W.); 2Laboratory of Metrology and Calibration, Beijing 100876, China; xywang0130@163.com (X.W.); liujuncheng@buaa.edu.cn (J.L.)

**Keywords:** miniaturization, timing system, radiation enhancement, broadband, long-distance communications

## Abstract

Precise timing synchronization remains a fundamental requirement for modern navigation and communication systems, where the miniaturization of Loran-C infrastructure presents both technical challenges and practical significance. Conventional miniaturized loop antennas cannot simultaneously meet the requirements of the Loran-C signal for both radiation intensity and bandwidth due to inherent quality factor (Q) limitations. A sub-cubic-meter impedance matching (IM) antenna is proposed, featuring a −20 dB bandwidth of 18 kHz and over 7-fold radiation enhancement. The proposed design leverages a planar-transformer-based impedance matching network to enable efficient 100 kHz operation in a compact form factor, while a resonant coil structure is adopted at the receiver side to enhance the system’s sensitivity. The miniaturized Loran-C timing system incorporating the IM antenna achieves an extended decoding range of >100 m with merely 100 W input power, exceeding conventional loop antennas limited to 30 m operation. This design successfully achieves overall miniaturization of the Loran-C timing system while breaking through the current transmission distance limitations of compact antennas, extending the effective transmission range to the hundred-meter scale. The design provides a case for developing compact yet high-performance Loran-C systems.

## 1. Introduction

Long-range navigation (Loran) is the most widely used land-based radio navigation system in the world, and the Loran-C system is one of them [1,2]. The Loran-C signal, operating at 100 kHz, has been widely used in navigation, positioning, and timing (PNT) as well as scientific research such as ground wave propagation, underwater military communications, and satellite/Loran-C integrated navigation [3,4]. Additionally, it can provide high-precision time and frequency signals in the order of hundreds of nanoseconds [5]. For the Loran-C system, the signal should be transmitted and received through the antenna. At present, the electric antenna is widely used in the Loran-C system [6]. However, since the length of the electric antenna is directly proportional to the wavelength, it will lead to a large volume and is vulnerable to power line interference and electrostatic deposition [7,8]. Consequently, the low-frequency antennas currently in use are restricted to deployment in unobstructed locations and are unsuitable for implementation in mobile equipment.

To realize the miniaturization of low-frequency antennas, mechanically actuated antennas were initially introduced. These systems utilize mechanically driven permanent magnets or electrets to produce periodic oscillations of magnetic and electric dipoles, enabling electromagnetic wave emission [9,10,11,12]. Owing to constraints in operational frequency and bandwidth, these antennas are incapable of fulfilling the demands for long-range Loran-C signal propagation. To facilitate the miniaturization of Loran-C systems, initial research efforts have focused exclusively on the receiving antennas [13]. D.L. et al. proposed to use a loop magnetic antenna instead of a whip antenna for the reception of the Loran-C signal, and verified that a small magnetic antenna has a similar acceptance ability to a 9-inch whip antenna [14]. K.W. et al. enhanced the reception capability of the magnetic antenna by incorporating an active filter circuit into the design [15]. While receiving antennas have achieved compact dimensions of merely decimeters to centimeters, Loran transmitter antennas persist at the hectometer scale owing to fundamental limitations in power handling capacity and operating bandwidth [16]. Consequently, the miniaturization of Loran-C transmitting antennas remains an urgent challenge to be addressed.

Small loop antennas have been validated as an effective approach for antenna miniaturization, yet their performance is fundamentally constrained by the quality factor Q, rendering them incapable of meeting both the bandwidth and radiation intensity requirements of Loran-C signals [17,18,19]. In previous studies, an isolation transformer-based low-frequency antenna successfully reduced the size of low-frequency transmitting antennas to the meter scale. By incorporating capacitive tuning to adjust the loop antenna’s inductive reactance, the radiation efficiency can be significantly improved [20]. The antenna’s frequency response demonstrates a bandpass characteristic similar to that of Loran-C signals. Building upon this structure, proper adjustment of specific physical parameters shows promise for achieving the miniaturization of Loran-C transmitting antennas.

This study demonstrates a miniaturized Loran-C timing system that reduces size to 1% of conventional implementations through synergistic integration of impedance matching (IM) and magnetic coupling (MC) enhancement. The transmitter employs a planar-structure transformer as an impedance matching network for the miniaturized loop antenna that provides 7-fold signal amplification while maintaining an 18 kHz bandwidth, overcoming the fundamental Q-factor limitation that has plagued traditional miniaturized antennas. The receiver’s optimized LC resonant coil demonstrates dual functionality as both a bandpass filter and signal amplifier, achieving two orders of magnitude magnetic field enhancement for raised reception quality. This integrated approach delivers a stable Loran-C signal transmission with mega-current capacity and effective noise suppression. Notably, reliable time synchronization over 100 m distances- a 3-fold improvement over conventional miniaturized systems. Field tests confirm mid-range transmission capability, establishing a new paradigm for deployable Loran-C systems in space-constrained applications without compromising the precision timing requirements characteristic of traditional large-scale implementations. The paper is organized as follows: Section 2 introduces the constructed system, along with its theoretical foundation and simulation results. Section 3 presents the experimental system setup and the demonstration of experimental results. Section 4 concludes the work and discusses future directions.

## 2. Theory and Simulation

Figure 1a,b illustrate the frequency and time-domain characteristics of standard Loran-C signals, defining the baseline antenna requirements. The design must support Binary Phase-Shift Keying (BPSK) modulation and exhibit a −20 dB bandwidth of about 20 kHz to match Loran-C’s spectral profile. Current research on 100 kHz miniaturized antennas demonstrates that loop antennas have been widely investigated due to their structural simplicity and tunable parameters. However, these antennas face fundamental limitations imposed by their quality factor (Q). As illustrated in Figure 1c, a multi-turn loop antenna exhibits improved gain but suffers from severely reduced bandwidth (the grey dotted line). Conversely, single-turn loop antennas maintain broader operational bandwidth but significantly compromise power handling capacity and radiation efficiency (the red short dotted line). In practical Loran-C signal transmission applications, this trade-off presents critical operational constraints: multi-turn loop antennas with insufficient bandwidth introduce severe signal distortion due to pronounced tailing effects. Single-turn implementations exhibit inadequate radiation intensity, making them susceptible to environmental noise interference, as shown in Figure 1d. The results underscore that neither conventional multi-turn nor single-turn configurations alone can satisfy the simultaneous requirements of bandwidth, radiation efficiency, and signal fidelity characteristic of Loran-C timing systems.

To overcome these inherent limitations while achieving system miniaturization, the proposed Loran-C timing system operates on the principle of near-field magnetic coupling, where the modulated Loran-C magnetic signal is radiated by the transmitting coil and induces detectable current in the receiving coil, as illustrated in Figure 2. Unlike conventional magnetic induction systems, this design incorporates a transformer-based impedance matching network at the transmitter to enable efficient high-current excitation of the radiating coil, enhancing the magnetic field intensity by over an order of magnitude. At the receiver side, a resonant coupling loop provides simultaneous bandpass filtering and signal amplification, improving both signal quality and operational distance. The system comprises a transmitter section with a BeiDou long-range precision Loran (BPL) signal generator producing accurate Loran-C waveforms that are amplified and fed into the impedance-matched antenna, and a receiver section where the coupled magnetic signals are captured and processed by a precision long-wave timing receiver for time information decoding. This optimized near-field coupling approach achieves substantial size reduction while preserving the critical timing synchronization capabilities characteristic of Loran-C systems, demonstrating the feasibility of miniaturized implementations without compromising performance.

The operating principle of the miniaturized Loran-C transceiver system can be explained through RLC circuit resonance and mutual inductance between inductors, as illustrated in Figure 3. In the transmitter circuit, a capacitor is connected in series with the transmitting coil, forming an RLC resonant circuit comprising the coil inductance, capacitance, and inherent resistance. At the resonant frequency, the circuit achieves minimum impedance equal to the wire’s internal resistance *R*_1_ (typically 10^−2^ Ω). The signal source and power amplifier are modeled as an AC source with input voltage *U*_in_ and input impedance *R*_in_ (typically 2.5–50 Ω), providing current to the transmitting coil. However, the significant impedance mismatch between *R*_1_ and Rin prevents efficient power transfer to the coil, substantially limiting radiation capability. To address this limitation, a transformer is employed as an impedance matching device, which can be represented as:(1)$$R_{\mathrm{in}}={\left(N_{p}/N_{s}\right)}^{2}\times R_{1}$$
where *N*_p_ and *N*_s_ represent primary and secondary winding turns, respectively, *I*_0_ denotes the primary current, and *I*_1_ indicates the secondary current. The transformer amplifies the coil current by a factor of *N*_p_/*N*_s_, proportionally enhancing the antenna’s radiation intensity. At this moment, the radiation intensity *B*_T_ can be formulated as [19]:(2)$$B_{T}=\frac{r^{2}\mu_{0}(N_{p}/N_{s})I}{2{\left(x^{2}+r^{3}\right)}^{3/2}}$$
where *μ*_0_ is permeability, *I* is current in the transmitting coil, and *x* is axial distance. The magnetic field generated by the transmitting antenna decays with the cube of distance (*r*^3^); as a result, a resonant coil is implemented at the receiver to enhance magnetic field intensity. The enhanced field strength *M*_2_ relates to the original field *M*_1_ through magnetic coupling [21,22,23]:(3)$$M_{2}=KM_{1}=\frac{\mu_{0}\pi r_{r}\omega N_{r}^{2}}{2Z_{r}}M_{1}$$
where *r*_r_ is the radius, *N*_r_ is the number of turns, *ω* is the resonance frequency, and *Z*_r_ is the impedance of the resonance coil. Since *K* must be greater than 1, it follows that *M*_2_ > *M*_1_. The magnetic field enhancement capability is primarily determined by the coil’s physical dimensions and number of turns. This enhanced field *M*_2_ induces current in the receiver coil *L*_R_, which is then processed by the BPL timing receiver. According to Lenz’s law, the enhanced magnetic field *M*_2_ induces a greater electromotive force in the receiving antenna, thereby facilitating signal decoding.

The unique spectral characteristics of Loran-C signals necessitate a balanced design approach that simultaneously optimizes antenna bandwidth and radiation efficiency. The antenna bandwidth is primarily governed by its quality factor (Q), expressed as [24]:(4)$$Q=\frac{1}{R_{1}}\sqrt{\frac{L_{T}}{C_{T}}}$$
where *L*_T_ represents the inductance of the transmitting antenna, and *C*_T_ denotes the series capacitance in the circuit. The *Q* is strongly influenced by the ratio of the series capacitance to the coil inductance—higher inductance leads to an increased *Q* and consequently narrower bandwidth. The inductance *L*_T_ of the coil can be approximated by [25]:(5)$$L_{T}=\frac{\mu_{0}N_{T}^{2}\pi r_{T}^{2}}{l}$$
where *r*_T_ is the radius, *N*_T_ is the number of turns, and *l* is the coil length of the transmitting coil. When *r*_T_ ≫ *l*, the inductance scales quadratically with the number of turns (*N*_T_^2^) and linearly with the radius (*r*_T_). Therefore, reducing either the number of turns or the coil radius effectively lowers the Q-factor, thereby broadening the antenna bandwidth. As expressed in (2), when the observation distance satisfies *x* ≫ *r*_T_, the radiation intensity *B*_T_ scales quadratically with the coil radius *r*_T_ but only linearly with the number of turns *N*_T_. This nonlinear relationship suggests that reducing the number of turns presents a more favorable approach for bandwidth enhancement, as it incurs a comparatively smaller penalty on radiation performance.

To systematically investigate the impact of various parameters on system transmission performance, comprehensive simulations of both transmitting and receiving antennas were conducted. Initial simulations examined antennas with identical inductance parameters but varying physical dimensions, as shown in Figure 4a. Due to their equivalent Q-factor characteristics, all configurations exhibited similar bandwidth performance. However, single-turn loop antennas demonstrated superior radiation efficiency, generating a magnetic field strength of approximately 1.6 mT, significantly higher than the sub-0.5 mT output of multi-turn configurations. As shown in Figure 4b, increasing the radius of single-turn loop antennas resulted in enhanced magnetic field radiation, though this improvement was accompanied by an increase in inductance and consequent bandwidth reduction. For instance, a 0.6 m radius antenna provided approximately 25% greater field strength compared to a 0.5 m version, at the cost of a 20% decrease in bandwidth of −3 dB. Notably, both configurations satisfied Loran-C transmission requirements with bandwidths of −20 dB exceeding 20 kHz.

The receiving subsystem employed a 0.3 m commercial ultra-wideband loop antenna with a flat 90–110 kHz response. Simulations revealed critical reception parameters as shown in Figure 4c,d. Resonant loops enhanced signal strength non-uniformly, peaking at 100 kHz and tapering toward band edges. While five-turn loops achieved strong amplification, their narrow 2 kHz bandwidth caused problematic signal tailing, as shown in Figure 4c, the gray shadow. Three-turn configurations provided optimal balance, delivering 100-fold amplification with 5 kHz bandwidth (the pink shadow). Dimensional analysis showed maximum performance when resonant loops matched the receiver’s 0.3 m size, as the concentrated near-field flux and Lenz’s law effects maximized voltage induction. These findings guided the final design selection of a 0.3 m, three-turn resonant loop that optimally balanced amplification and bandwidth requirements.

## 3. Experiment and Results

Based on the simulation results, antenna prototypes were fabricated as shown in Figure 5. The transmitting coil was designed as a 1 m × 1 m square loop, exhibiting similar electromagnetic characteristics to a circular loop with 0.56 m radius. The measured inductance of the antenna was approximately 3.91 μH, matched with 648 nF capacitors for optimal resonance. The resonant point impedance of the antenna is approximately 0.05 Ω, requiring impedance matching to interface with the power amplifier’s 2.5 Ω output. To achieve this, an impedance-matching transformer with a turns ratio of 7:1 (Np:Ns) was implemented. For magnetic field enhancement, a 3-turn resonant loop with 30 cm radius was constructed using 150 nF tuning capacitors according to simulation. An outdoor testbed was established to evaluate the antenna performance. The transmission system consisted of: A function generator producing standard Loran-C pulse signals, an ATA-3090C power amplifier (1000 W output power, 20 Ap current, 50 Vp voltage), and an oscilloscope monitoring the antenna’s input voltage and current. The receiving system employed: a miniaturized active loop antenna (0.3 m radius), an oscilloscope capturing the induced voltage.

Experimental results demonstrate the significant impact of transformer selection on the IM antenna’s radiation performance. Spectral analyzer measurements reveal the superior performance of planar transformers, which not only enhance radiation intensity and reduce energy loss at equivalent power levels but also expand the antenna bandwidth to 1.5 times that of conventional transformers due to their reduced inductance, as shown in Figure 6a. The blue line represents the −20 dB reference line. The black dashed line indicates the frequency spectrum of the antenna using a planar transformer (with gray lines marking its bandwidth), while the red dashed line shows the frequency spectrum of the antenna employing a conventional transformer (with pink lines denoting its bandwidth). Comparative analysis in Figure 6b shows that the IM antenna achieves over 3 dBm higher radiation intensity than conventional multi-turn loop antennas while retaining the bandwidth advantage of single-turn designs, maintaining a bandwidth of −20 dB exceeding 18 kHz. The black line represents the proposed antenna, the blue line represents the traditional multi-turn loop antenna, and the red represents the traditional single-turn loop antenna. The planar transformer implementation effectively addresses the fundamental trade-off between radiation efficiency and bandwidth that has traditionally limited Loran-C antenna performance, demonstrating clear advantages for timing signal transmission applications where both parameters are critical. Comparative current waveform measurements at 10 m distance (Figure 6c,d) demonstrated excellent signal fidelity-the received pulses maintained precise temporal characteristics without observable distortion, with clearly distinguishable inter-pulse intervals suitable for immediate acquisition and decoding by BPL timing receivers.

To validate long-range timing performance, a complete miniaturized Loran-C system was tested in open-field conditions as depicted in Figure 7. The BPL signal simulator encoded the date “2008.05.12” by using the Loran-C signal, while the long-wave timing receiver was initialized to “2000.01.01”. Successful time transfer was confirmed when the receiver synchronized to the transmitted date. At a transmission distance of 100 m, the system demonstrated excellent performance characteristics: it achieved rapid signal acquisition within seconds (faster than conventional miniaturized antenna transmissions), maintained completely error-free data decoding, and exhibited stable long-term synchronization capability throughout the testing period. Notably, the power amplifier in this configuration delivers approximately 100 W of output power. This performance represents a significant improvement over conventional alternatives: under identical power conditions, a similarly sized loop antenna without IM or MC technology achieves only 30 m effective decoding distance. The implementation of IM alone extends this range to 60 m, while the complete system with IM and MC technology exceeds 100 m, delivering a 3-fold enhancement in transmission distance compared to traditional designs. According to Equation (2), higher power input may enable additional extension of the timing signal’s effective range. These results collectively validate the system’s operational reliability and superior timing performance compared to traditional implementations. These experimental results verify that the miniaturized design maintains all critical Loran-C timing functionalities while dramatically reducing physical dimensions.

## 4. Conclusions

This study presents a miniaturized Loran-C timing system that successfully integrates high-power capacity, compact design, and long-range transmission capabilities. The IM antenna, designed with a planar transformer and operating at 90–110 kHz, features a compact structure of less than 1 m^3^, a −20 dB bandwidth exceeding 18 kHz, kilowatt-level power capacity, and millitesla-scale magnetic field radiation intensity. Outdoor tests have validated the Loran-C timing capability of the system, achieving rapid signal acquisition within seconds and stable synchronization over distances exceeding 100 m with merely 100 W input power. The experimental results align closely with theoretical predictions and simulations, confirming the effectiveness of the proposed design. This work provides a practical solution for deploying compact yet high-performance Loran-C systems in space-constrained applications, paving the way for future advancements in portable navigation and timing technologies.

## Figures and Tables

**Figure 1 sensors-25-04352-f001:**
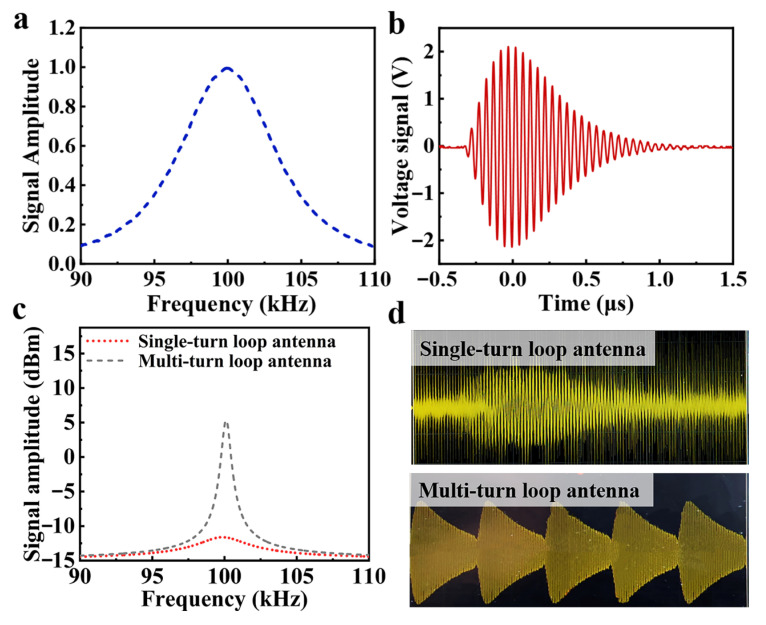
(**a**) Spectrum and (**b**) time domain diagram of standard Loran-C signal. (**c**) Radiation intensity and (**d**) oscillogram of traditional loop antenna.

**Figure 2 sensors-25-04352-f002:**
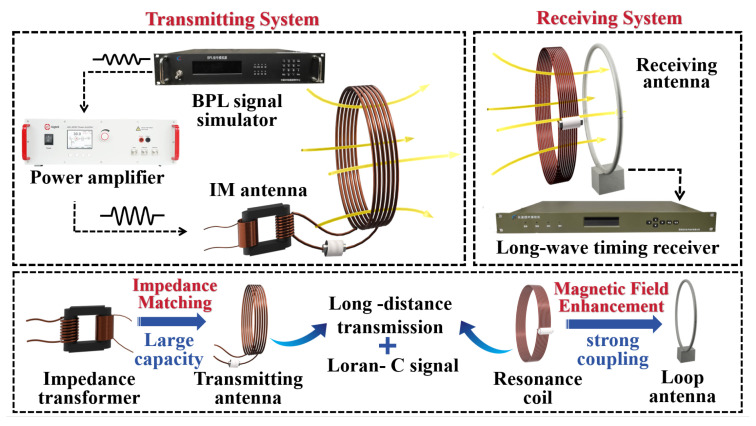
Miniaturized Loran-C timing system.

**Figure 3 sensors-25-04352-f003:**
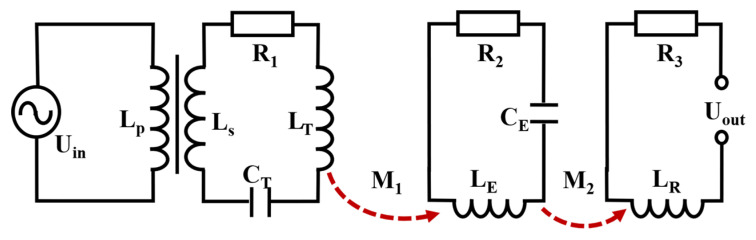
Equivalent circuit model of miniaturized Loran-C timing system.

**Figure 4 sensors-25-04352-f004:**
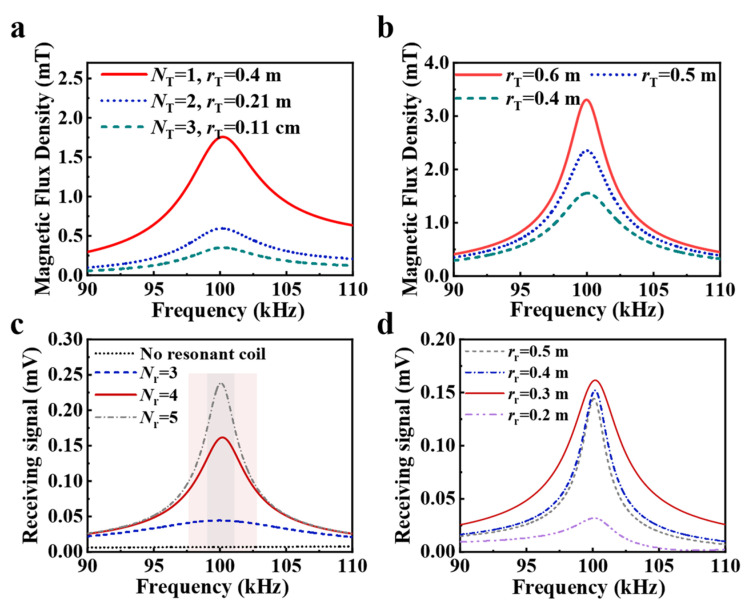
Magnetic flux density of the transmitting coil with different (**a**) number of turns and (**b**) radius. And receiving signal with different (**c**) turns and (**d**) radius of resonant coil.

**Figure 5 sensors-25-04352-f005:**
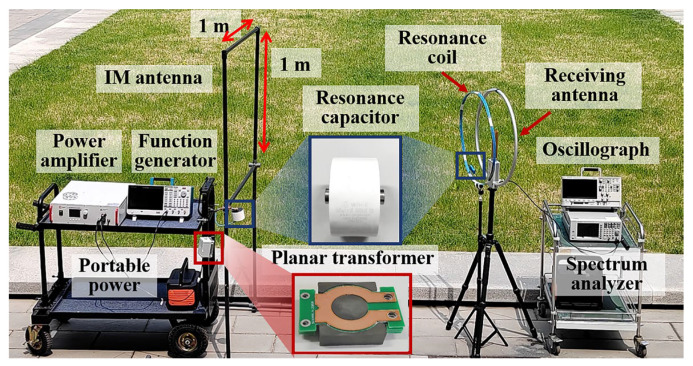
Experimental setup of the antenna test system.

**Figure 6 sensors-25-04352-f006:**
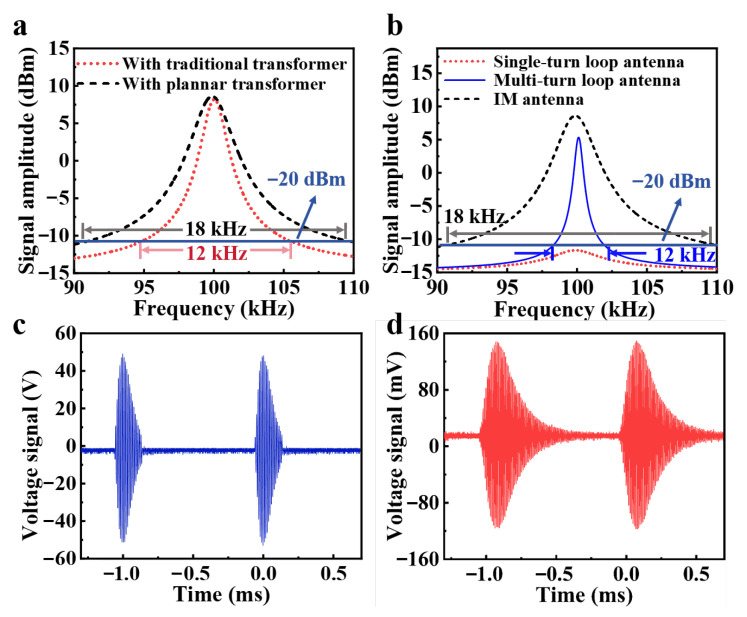
(**a**) The signal spectrum of the IM antenna using different transformers. (**b**) Signal spectrum of the IM antenna and the traditional loop antenna. Comparison of (**c**) transmitting signal and (**d**) receiving signal.

**Figure 7 sensors-25-04352-f007:**
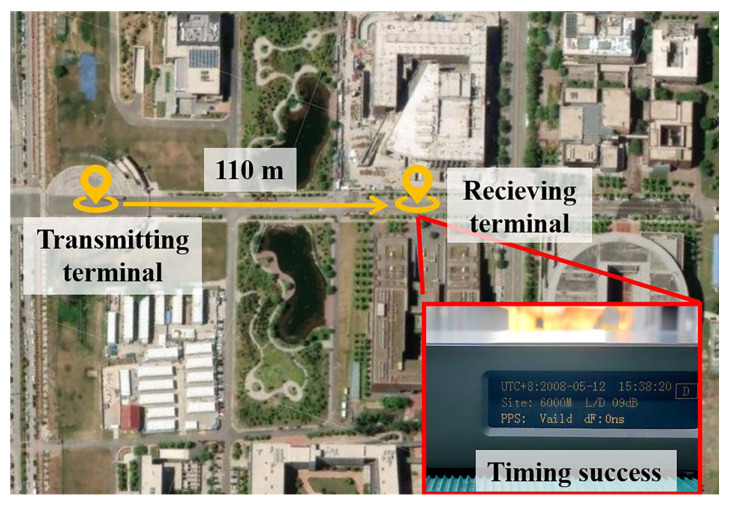
Outdoor long-distance decoding test experiment panorama.

## Data Availability

The original contributions presented in this study are included in the article.

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
