# Peer review of "High-Power Coupled Wideband Low-Frequency Antenna Design for Enhanced Long-Range Loran-C Timing Synchronization"

_sensors, 2025, doi:10.3390/s25144352_

Round 1

Reviewer 1 Report

Comments and Suggestions for Authors

- The paper provides a practical, validated solution for deploying Loran-C technology in modern compact environments showing the significant miniaturization using a planar-structure transformer to achieve long range with low power.  The authors should clearly state the main novelty and point out the main advantage of this proposed design versus the traditional system in terms of which challenges are met to bridge the gap between the traditional and this efficient design.
The paper is well written and organized. I would suggest the following to improve the paper:

  1. Abstract: please provide full name for the abbreviation IM.
  2. In the last paragraph of the introduction please be specific if this is a new proposal or modified. If latter, please reference it correctly.
  3. Please provide an outline of the paper at the end of the Introduction.
  4. Is it possible to enlarge all figures?
  5. Fig. 2 is not clearly visible in black and white when printed - please, correct it.
  6. If it is possible, the authors should add a table with performance comparison which would provide a great insight into the benefits.

    7. Is it possible to include time deviation to quantify synchronization accuracy across distances and under varying environmental conditions?

    8.  Fig. 6 is not very visible in black and white.  I would be better if it is possible to enlarge them and mark more clearly the bandwidth on a) and b) plots - maybe by putting a horizontal line?

    9. Related to the magnetic field distribution, did the authors looked into the field strength plots (due to the safety issues)?

    10. Are there any limitations - related to the range constraints or interference?

Reviewer 2 Report

Comments and Suggestions for Authors

This paper presents a significant technical advancement in creating compact, high-performance Loran‑C systems suitable for mobile, space-constrained applications. However, several questions should be fixed as shown below:

  1. Though ~18 kHz is near Loran‑C’s -20 dB target (~20 kHz), it's still slightly deficient and below –3 dB benchmarks. Can this support full BPSK decoding under real-world noise?
  2. Field tests achieved 100 m with 100 W, but scaling to kilowatt levels to reach kilometre range is suggested theoretically only—no empirical data.
Comments on the Quality of English Language

The writing is generally clear and professional, but a few long, complex sentences, occasionally hinder readability, which can be improved

Reviewer 3 Report

Comments and Suggestions for Authors

While the manuscript claims a “>100 m” decoding range with only 100 W input power, it lacks a rigorous comparison with state-of-the-art compact Loran-C antennas or related sub-100 kHz systems. Please provide a detailed performance benchmark—including bandwidth, gain, efficiency, and size metrics—against peer designs to substantiate the claimed "10-fold radiation enhancement."

The paper frequently references "radiation enhancement" without sufficiently explaining the physical mechanisms or simulation assumptions behind this improvement. Is the enhancement due to near-field coupling, matching efficiency, or actual far-field gain? Detailed electromagnetic modeling results and field plots must be included to validate the claims.

The -20 dB bandwidth of 18 kHz is mentioned as a key achievement. However, the paper does not explain how this translates into system-level timing accuracy or stability, which are the ultimate metrics in a Loran-C context. How does this bandwidth relate to the Group Delay Flatness or timing jitter at 100 kHz?

The validation appears limited and lacks rigorous environmental testing, sensitivity analysis, or frequency stability measurements over temperature and load variations. A meaningful field test over varied terrain or urban environments would greatly enhance the paper's credibility.

The planar-transformer-based impedance matching network is central to the miniaturization claim but is insufficiently described. The manuscript must provide detailed schematics, matching efficiency plots, and a discussion of parasitic effects at 100 kHz.

Phrases like “breakthrough,” “significantly outperforming,” and “framework for development” are subjective and should be replaced with objective, quantifiable statements backed by data or simulation results.

A 100 m decoding range may be impressive in lab settings, but real-world applicability requires discussion on regulatory constraints (e.g., FCC or ITU emission limits at 100 kHz), electromagnetic compatibility (EMC), and antenna orientation or grounding requirements.

Reviewer 4 Report

Comments and Suggestions for Authors

This paper proposes a high-power coupled wideband low-frequency antenna design to improve long-range timing synchronization performance in Loran-C systems. The topic is practically valuable for precision navigation and legacy timing infrastructure, but the manuscript suffers from issues.

  1. While the paper claims to improve Loran-C timing, it does not: Quantify jitter, propagation delay stability, or time drift improvements, and the paper does not compare with other timing standards (e.g., GNSS, IEEE 1588).
  2. Introduce quantitative metrics for timing synchronization impact (e.g., nanosecond-level variation), possibly via channel modeling or signal recovery simulation. Low Complexity MIMO-FBMC Sparse Channel Parameter Estimation for Industrial Big Data Communications, MBPD: A Robust Algorithm for Polar-Domain Channel Estimation in Near-Field Wideband XL-MIMO Systems.
  3. It is suggested to Include an S-parameter matrix (S21) comparison before and after optimization.
  4. No uncertainty or measurement confidence is discussed.

Round 2

Reviewer 3 Report

Comments and Suggestions for Authors

The authors have addressed all my comments.